# Active Site Engineering on Two-Dimensional-Layered Transition Metal Dichalcogenides for Electrochemical Energy Applications: A Mini-Review

Chueh-An Chen [1], Chiao-Lin Lee [1], Po-Kang Yang [2], Dung-Sheng Tsai [3,*] and Chuan-Pei Lee [1,*]

[1] Department of Applied Physics and Chemistry, University of Taipei, Taipei 10048, Taiwan; giant851229@gmail.com (C.-A.C.); lee.annie.lin.9@gmail.com (C.-L.L.)

[2] Graduate Institute of Nanomedicine and Medical Engineering, College of Biomedical Engineering, Taipei Medical University, Taipei 11031, Taiwan; yangpk@tmu.edu.tw

[3] Department of Electronic Engineering, Chung Yuan Christian University, Taoyuan City 32023, Taiwan

* Correspondence: dungsheng@cycu.edu.tw (D.-S.T.); CPLee@utaipei.edu.tw (C.-P.L.)

**Abstract:** Two-dimensional-layered transition metal dichalcogenides (2D-layered TMDs) are a chemically diverse class of compounds having variable band gaps and remarkable electrochemical properties, which make them potential materials for applications in the field of electrochemical energy. To date, 2D-layered TMDs have been wildly used in water-splitting systems, dye-sensitized solar cells, supercapacitors, and some catalysis systems, etc., and the pertinent devices exhibit good performances. However, several reports have also indicated that the active sites for catalytic reaction are mainly located on the edge sites of 2D-layered TMDs, and their basal plane shows poor activity toward catalysis reaction. Accordingly, many studies have reported various approaches, namely active-site engineering, to address this issue, including plasma treatment, edge site formation, heteroatom-doping, nano-sized TMD pieces, highly curved structures, and surface modification via nano-sized catalyst decoration, etc. In this article, we provide a short review for the active-site engineering on 2D-layered TMDs and their applications in electrochemical energy. Finally, the future perspectives for 2D-layered TMD catalysts will also be briefly discussed.

**Keywords:** active site; catalysis; electrochemical energy; transition metal dichalcogenides; two-dimensional materials

## 1. Introduction

Since the exfoliation of graphene in 2004, 2D-layered materials have caused global research fanaticism based on its unique property [1–7]. In recent years, the layered transition metal dichalcogenides (TMDs) have been regarded as a hot topic in the research of 2D-layered materials due to their low cost, abundance, and easy processing [8]. These materials also show great potential in many fields due to the distinct physical and chemical properties [9–16]. Unlike bulk materials, the performance of 2D-layered TMDs is determined by the thickness, band gap, interface, interlayer distance, and surface structure [14,17–21]. Moreover, the proper modification will greatly enhance the performance of these materials. Therefore, people are committed to finding the most effective modification method, such as curvature structure, plasma treatment, doping, and surface modification [22–24]. Recently, Zeng et al. showed that the band gap of two-dimensional materials could be manipulated via spherical diameter engineering (SDE) technology. By the construction of negative or positive curvature, the increase and decrease in the band gap resulting from compressive and tensile lattice deformation reveals that the optical property of 2D-layered materials can be modified by curvature engineering [25]. In addition, an effective strategy has also been proposed for the doping of $MoS_2$ with nitrogen through a remote $N_2$ plasma surface treatment [26]. Furthermore, a recent report revealed that the contact resistance (Rc) in the few-layered $WS_2$ and $MoS_2$ can be reduced greatly with chloride molecular

doping, attributing that the Schottky barrier width decreases with the high electron-doping density [27].

Although the surface of 2D-layered materials with minimal roughness and dangling bonds can be regarded as basal planes, which are useful for electrical applications, the catalytic activity of 2D-layered material basal planes is poor. The catalytic activity is basically located at the edge sites of 2D-layered materials because more dangling bonds can trigger more catalysis [28,29]. Therefore, a new strategy has been proposed to achieve controllable defect engineering in $MoS_2$ ultrathin nanosheets for creating more active edge sites [30]. Besides, the surface modulation has also been developed for the effective decoration of isolated Ni atoms on hierarchical $MoS_2$ nanosheets supported on multichannel carbon matrix nanofibers, showing highly active and stable hydrogen evolution reaction (HER) [31]. Above-mentioned reports have demonstrated that proper modification or treatment can effectively improve the catalytic performance of 2D-layered materials. In this article, we provide a short review of the active-site engineering on 2D-layered TMDs and their applications in electrochemical energy; the various approaches will be briefly discussed as the following.

## 2. Curvature Engineering on 2D-Layered TMDs

The sphere diameter engineering (SDE) technique was demonstrated by Zeng et al. for manipulating the bandgap of 2D-layered materials [25]. By creating an isotropic curved surface, the strain of $MoS_2$ crystals can be induced [25]. After the $MoS_2$ growth processes via SDE, the liquid glass plane changes into a sphere having a specific diameter, resulting in a homogeneous and controllable lattice deformation of the supported 2D-layered materials with great precision and reliability. Moreover, the mass production of $MoS_2$ with a specific bandgap can be fabricated by controlling the glass amount becoming the glass sphere. As compared to traditional strategies (i.e., unidirectional strain engineering), it is worth noting that the higher tuning performance of this approach owing to the lattice deformation of the $MoS_2$ crystal is isotropic and multidirectional [18,32,33]. Moreover, this approach can obtain a uniform distribution of lattice deformation for all flakes on the entire isotropic sphere. The photoluminescence (PL) spectra of the $MoS_2$ crystals grown on the glass spheres having different diameters show that a peak redshift of the neutral exciton can be detected with the decrease in the diameter of the sphere due to the electronic band structure tuning resulting from the lattice deformation increase of the monolayer $MoS_2$ crystal [34]. By using this SDE strategy, the sphere design and assembly could boost the bandgap tuning of the post-growth $MoS_2$ and maintain the uniform lattice deformation of 2D-layered crystals over the surface of the isotropic sphere. This technique opens avenues to tune the bandgap of low-dimensional nanomaterials (such as 2D-layered materials and nanowires) and help promote their practical applications [25].

In 2018, Gonz'alez et al. investigated the bending process of 2D-layered materials, which is related to external forces [35], and studied the bending of a $MoS_2$ band in order to increase the complexity of the material, in this case, using the bending force on the central Mo layer while permitting the S atoms to move freely. As shown in Figure 1, the inner, bulk-like bands (the red lines) are qualitatively similar for R = 8.12 Å and R→∞, but the band gap is ~0.8 eV smaller, indicating that it is possible to control the bandgap of a $MoS_2$ nanosheet by bending it [35].

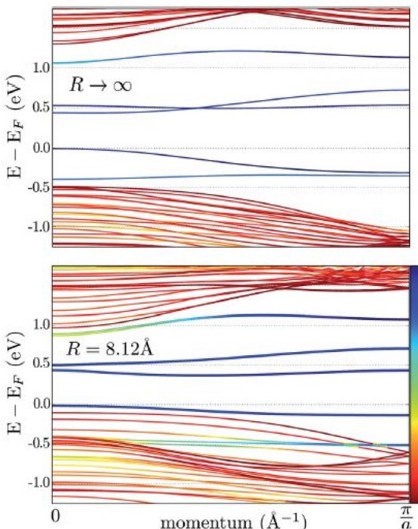

**Figure 1.** Band structure of a flat (upper panel) and a bent (lower panels) MoS$_2$ nanoribbon [35]. The different color shows the location of the eigenvalues: The inner subbands (i.e., bulk-like) and the edge subbands are red and blue, respectively; the hybridization of bulk and edge states are between red and blue.

Developing effective catalysts can be achieved by controlling the surface structure on the atomic scale; it is well known that the edge sites of a MoS$_2$ nanosheet are active for catalysis. Synthesizing a contiguous thin film consisting of highly ordered double-gyroid MoS$_2$ crystals having nanoscale pores can efficiently expose the edge sites of MoS$_2$. The high surface curvature of the mesoporous double-gyroid MoS$_2$ exposing large edge sites could provide excellent electrocatalytic activity toward hydrogen evolution reaction (HER) [36]. The HER activity data (i.e., cyclic voltammograms) of double-gyroid MoS$_2$ with Mo deposition times of 10 s, 20 s, and 1 min show that the increased surface area for samples deposited with longer deposition times is obvious when comparing the difference in capacitance current in the region of $-0.1$ to $+0.4$ V versus the reversible hydrogen electrode (RHE). The sample deposited for 1 min shows the greatest capacitance while the sample deposited for 10 s shows the smallest capacitance. Meanwhile, this double-gyroid MoS$_2$ also exhibited potential HER performance as compared to traditionally noble metal catalysts, such as Pt [36].

### 3. Plasma Treatment on 2D-Layered TMDs

A higher number of defects of pristine single-layer MoS$_2$ could be created via oxygen plasma exposure and hydrogen treatment, resulting in a high density of exposed edges and an obvious enhancement in the hydrogen evolution activity [37]. In Figure 2a, the intensity of both A$_{1g}$ and E$^1_{2g}$ modes of single-layer MoS$_2$ decreases with the oxygen plasma treatment time, indicating that the MoS$_2$ lattice distortion is caused by oxygen plasma [38]. In addition, a new peak at 285 cm$^{-1}$ (Mo$-$O bonds) becomes more obvious with the increase in the oxygen bombardment time [39]. Moreover, as shown in Figure 2b, the intensity of the PL peak (680 nm) also lowers with the increase in plasma treatment time, indicating that more defects and cracks were induced by the bombardment of oxygen [40,41]. Accordingly, oxygen plasma can not only decrease the MoS$_2$ crystal symmetry but also increase the lattice distortion, inducing more defects in MoS$_2$ for boosting electrochemical catalytic reactions. In Figure 3a, to reveal the effects of defects on the catalytic property of MoS$_2$, the HER property characterization of the MoS$_2$ monolayer with different oxygen plasma exposure times (0, 10, and 20 s) was measured. The sample (with 20 s oxygen plasma treatment) expresses the smallest onset potential, attributing to the largest amount of electrochemically active sites of MoS$_2$ after plasma exposure. Moreover, a positive correlation exists between the enhancement of HER activity and the plasma exposure time after the

correction of the $j-V$ curve for *iR* losses via MoS$_2$ itself. As shown in Figure 3b, the Tafel slope (~171 mV dec$^{-1}$) of MoS$_2$ with 20 s plasma treatment obviously decreases, indicating that the defects in the single-layer MoS$_2$ basal plane can be induced effectively by oxygen plasma exposure for boosting HER [37].

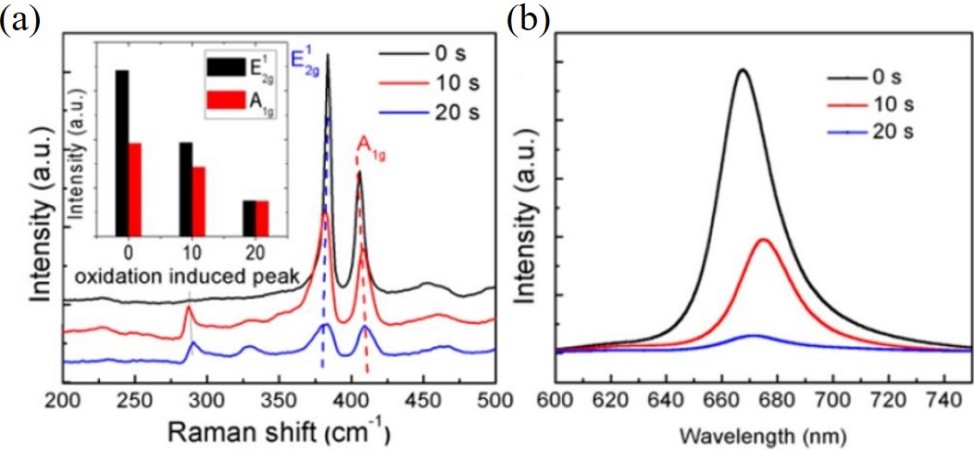

**Figure 2.** (**a**) Raman and (**b**) PL spectra of single-layer MoS$_2$ show that the peak intensity and position decrease and shift, respectively, with different oxygen plasma exposure times [37].

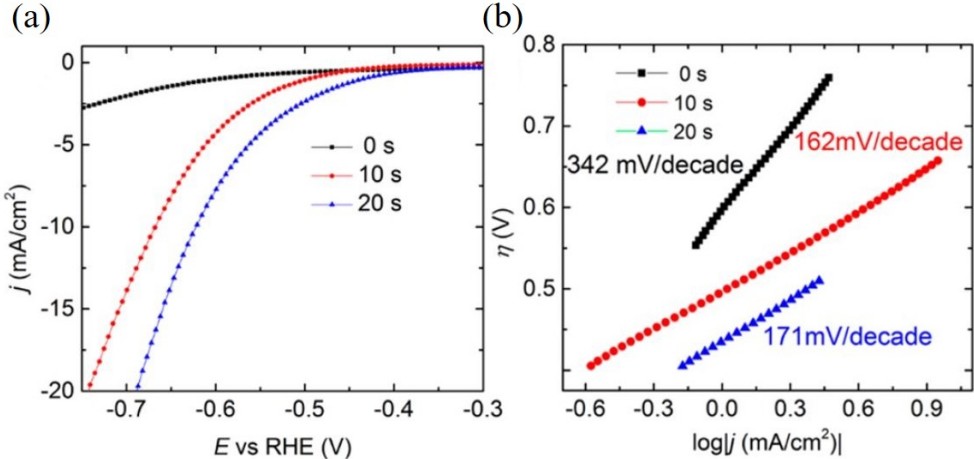

**Figure 3.** Hydrogen evolution reaction (HER) properties of single-layer MoS$_2$ with different oxygen plasma treatment times (0, 10, and 20 s). (**a**) Linear sweep voltammetry (LSV) curves show a positive correlation between the HER activity enhancement of single-layer MoS$_2$ and the plasma exposure time at the cathodic sweep of the first cycle. (**b**) Tafel plots show the better performance in electrochemical activity of the single-layer MoS$_2$ catalyst after 20 s oxygen plasma treatment [37].

In 2019, Nguyen et al. reported that the active sites can be effectively induced within the continuous MoS$_2$ sheets by plasma treatment [42]. Maximum active sites in large-area single-layer MoS$_2$ were obtained via N$_2$ plasma to meet the requirement of a high HER performance. In order to reveal the effect of N$_2$ plasma on the optical characteristics of MoS$_2$, the Raman and PL spectroscopy are shown in Figure 4. The Raman peak (E$_{2g}$ and A$_{1g}$) intensity and width of MoS$_2$ decrease a lot and become broadened, respectively, implying that a lot of lattice distortion and defects (such as cracks and sulfur vacancies) generated after plasma treatment [37]. On the other hand, the PL intensity also rapidly decreased with plasma treatment time, attributing to more defects site induced by N$_2$ plasma. For the HER performance, the MoS$_2$ without plasma treatment showed relatively low HER activities with an onset potential at −460 mV (vs. reversible hydrogen electrode (RHE)). By contrast, the samples with N$_2$ plasma treatment for 10–30 min showed a considerable decrease in

the onset potential. The smallest value of the onset potential ($-210$ mV) was exhibited by the 20 min plasma-treated $MoS_2$, as shown in Figure 5. Although this method is simple and controllable to maximize the HER efficiency of the wafer-scale $MoS_2$, the plasma treatment time should still be carefully controlled for optimizing the active site. This method holds promise for further improving the HER efficiency of 2D-layered $TMD_S$ [42].

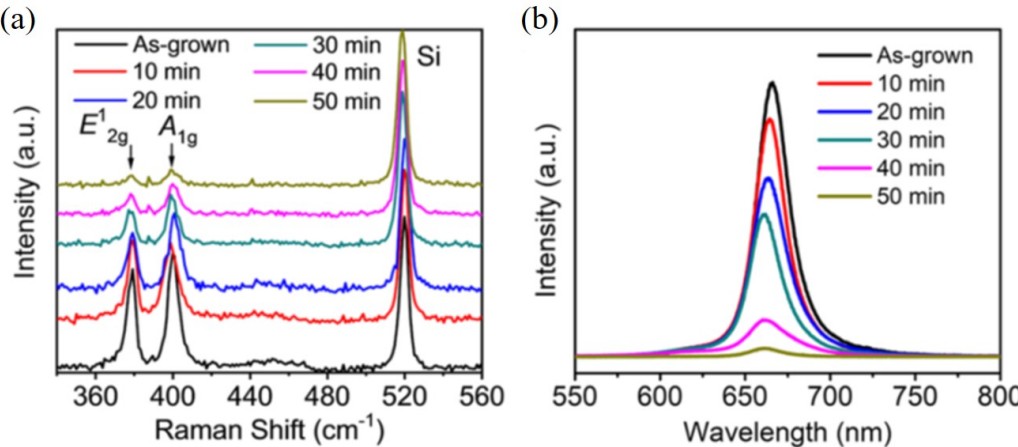

**Figure 4.** (**a**) Raman and (**b**) PL characteristics of $MoS_2$ with varying $N_2$ plasma treatment time [42].

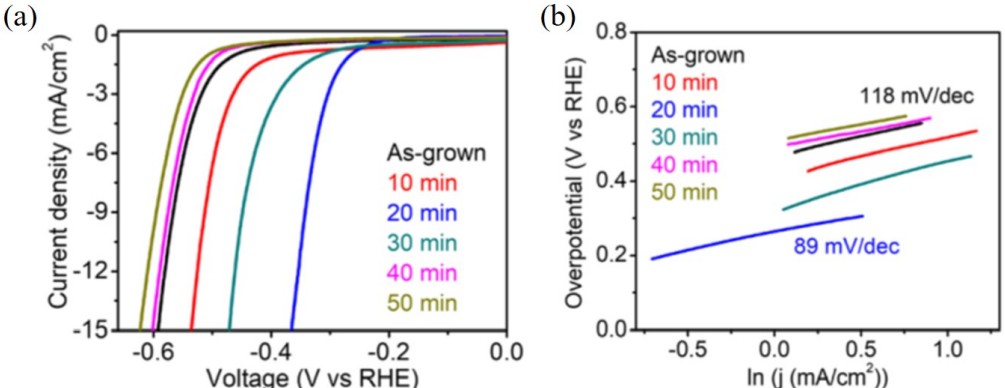

**Figure 5.** (**a**) iR-corrected linear sweep voltammograms and (**b**) Tafel plots with varying $N_2$ plasma treatment time [42].

Tao et al. reported that the electrocatalytic activity of $MoS_2$ thin-films for the HER significantly enhanced after using Ar or $O_2$ plasma treatment [43]. Figure 6 illustrates the processes of $O_2$ or Ar plasma treatment. With different Ar or $O_2$ plasma treatment durations, the electronic properties and defect sites of $MoS_2$ thin films could be adjusted. The polarization curves of the $MoS_2$ thin films before and after plasma treatment are shown in Figure 7a,b. To compare with the pristine $MoS_2$, the Ar and $O_2$ plasma-treated $MoS_2$ thin films show higher HER performance. At the beginning, the HER performance of $MoS_2$ was improved with increasing Ar plasma treatment time due to the increase in defects. After Ar plasma treatment for 720 s, the $MoS_2$ exhibited the best HER activity (a current density of 16.3 mA cm$^{-2}$ at $-350$ mV). However, the HER activity sharply declined (11.83 mA cm$^{-2}$) with increasing Ar plasma treatment time to 960 s, owing to the decrease in active species for too long a plasma treatment process. Furthermore, $MoS_2$ with $O_2$ plasma treatment showed a similar behavior of the HER activity. After 480 s $O_2$ plasma treatment, the maximum HER activity of $MoS_2$ was obtained. The HER activity decreased from 480 to 720 s plasma treatment, resulting from over-doping oxygen species, and the active species decreased for long-term plasma treatment. This report demonstrated that electrocatalytic activity on the 2D-layered material surface could be modified by controlling the plasma irradiation time [43]; however, over oxidation of $MoS_2$ could probably cause negative

effects on the device performance due to the decreases in active species. Moreover, it is well known that the formation of molybdenum oxides could also decrease the carrier mobility in pertinent electronic devices and result in the poor stability in acidic or alkaline electrolytes of the electrolysis system.

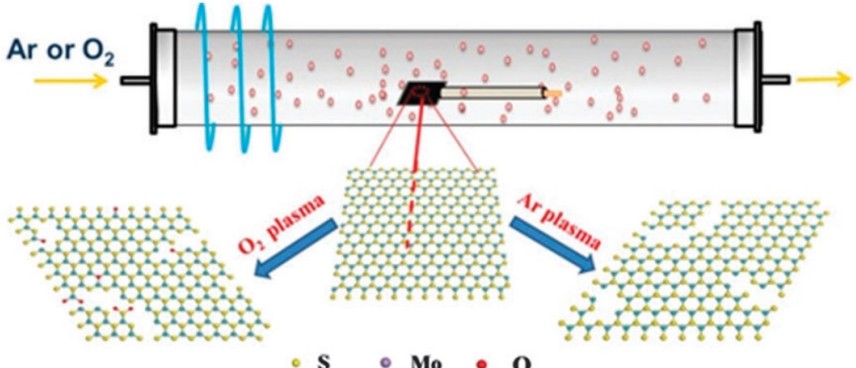

**Figure 6.** The development of defect sites in the $MoS_2$ thin films after $O_2$ or Ar plasma treatment [43].

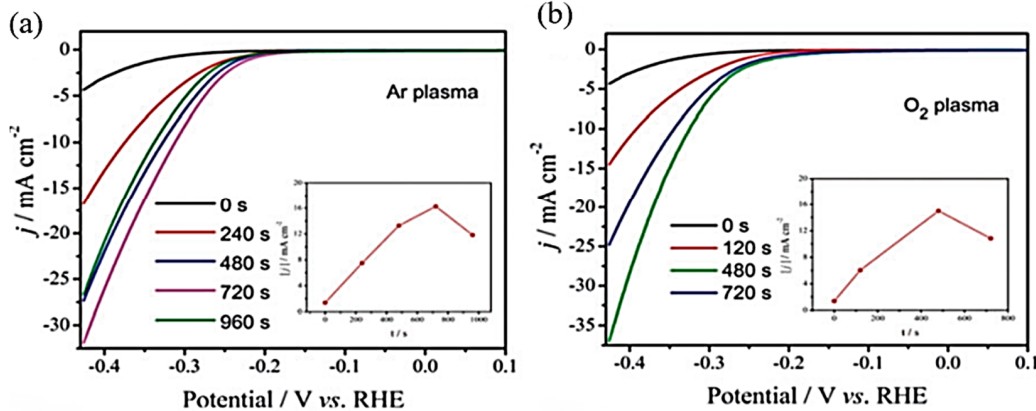

**Figure 7.** The polarization curves of the $MoS_2$ thin films at a 2 mV s$^{-1}$ scan rate after different (**a**) Ar plasma and (**b**) $O_2$ plasma treatment times [43].

## 4. Heteroatom-Doping on 2D-Layered TMDs

Shi et al. proposed a strategy to enhance the HER performance of $MoS_2$ by direct transition-metal doping [27]. The HER active centers of $MoS_2$ come from the edge deformed sites, as everyone knows [37]. Therefore, to enhance the HER catalytic activity of $MoS_2$, the effect of Zn atom doping on HER was investigated. The Raman spectrum of Zn-$MoS_2$ and the pure $MoS_2$ show a similar curve, as shown in Figure 8, indicating that there is no change in the $MoS_2$ layered feature of Zn-$MoS_2$. In other words, the physical characterizations show the high uniformity of Zn atom doping in the $MoS_2$ structure. As shown in Figure 9a (green line), the bulk $MoS_2$ shows poor HER activity (Tafel slope of 692 mV dec$^{-1}$) as in the previous report, owing to huge lateral resistance. To compare with pure $MoS_2$, Zn-$MoS_2$ exhibits superior electrocatalytic activity for HER, as shown in Figure 9a, promising to reduce the gap with the commercial Pt/C catalyst (Figure 9a, blue line). For the long-term stability test, the profile for the cathodic wave after 1000 cyclic potential sweeps shows a similar current density with the initial curve, as shown in Figure 9b, indicating the ultra-high stability of Zn-$MoS_2$ for HER. This low-cost strategy was demonstrated to design highly efficient electrocatalysts for HER, holding promise for broad applications in other catalytic systems [27].

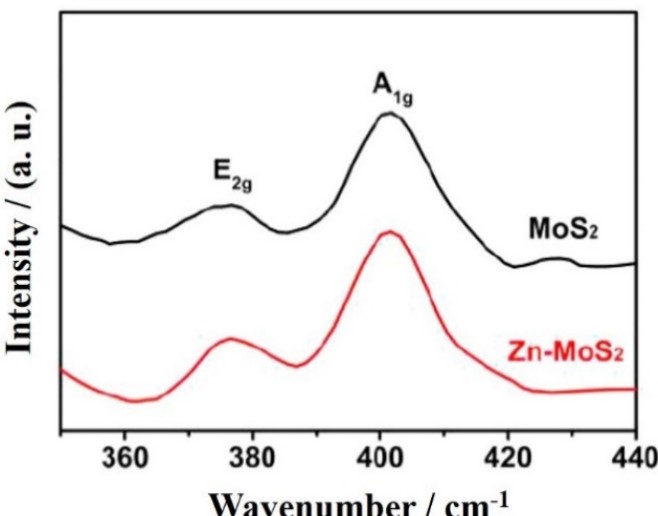

**Figure 8.** Raman characteristics of pure MoS$_2$ (black line) and Zn-MoS$_2$ (red line) [27].

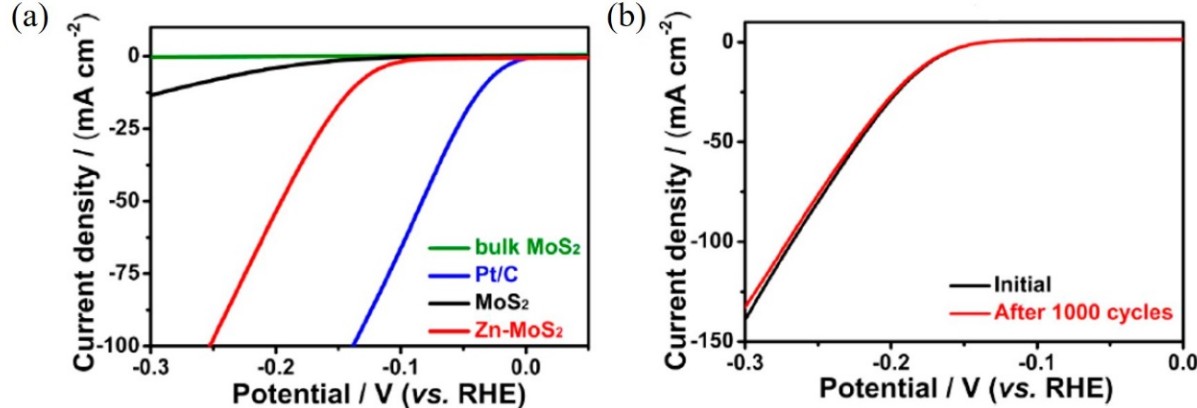

**Figure 9.** (**a**) Polarization curves of HER for bulk MoS$_2$, Pt/C, pure MoS$_2$, and Zn-MoS$_2$. (**b**) The polarization stability tests of Zn-MoS$_2$ catalyst (black line: Initial state; red line: After 1000 cycles) [27].

In 2015, Deng et al. reported that the in-plane S atom catalytic activity of MoS$_2$ can be activated with single-atom metal doping. As shown in Figure 10a, compared to the blank glassy carbon (GC) electrode and bulk MoS$_2$, the pure few-layer MoS$_2$ nanosheets (FL-MoS$_2$) without Pt atom doping showed a higher activity [44]. However, the activity of FL-MoS$_2$ without Pt atom doping was still much lower than with the 40% Pt/C catalyst. There is a big difference in the Tafel value of Pt–MoS$_2$ (96 mV dec$^{-1}$) and of the Pt/C electrocatalyst (32 mV dec$^{-1}$), as shown in Figure 10b, implying that the mechanism should be low relative to Pt. By contrast, the Tafel value of Pt–MoS$_2$ is closer to FL-MoS$_2$ (98 mV dec$^{-1}$). This result implies that the active sites of HER should also come from the S atoms rather than the Pt atoms in the Pt–MoS$_2$ sample. For stability testing, the Pt-MoS$_2$ catalyst exhibited a very stable performance after 5000 cycles of cyclic voltammetry (CV) scans, as shown in Figure 10c [44].

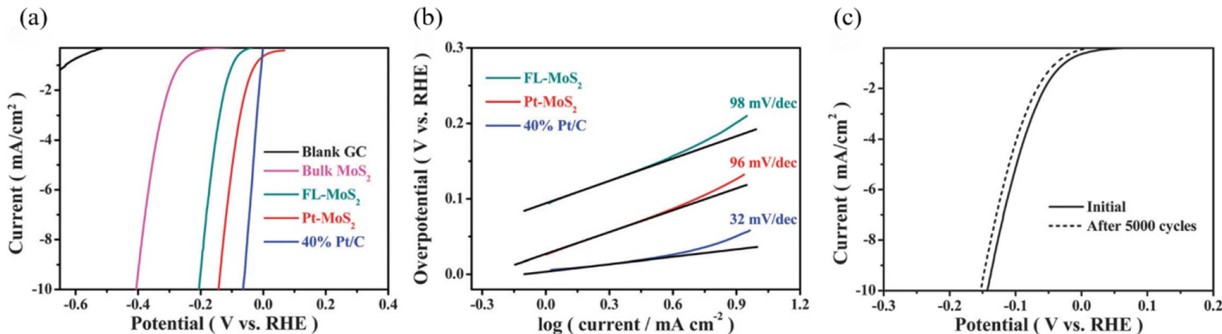

**Figure 10.** (**a**) HER polarization curves of Pt–MoS$_2$, blank glassy carbon (GC) electrode, bulk MoS$_2$, few-layer (FL)-MoS$_2$, and 40% Pt/C, respectively. (**b**) Tafel plots of FL-MoS$_2$ (98 mV dec$^{-1}$), Pt–MoS$_2$ (96 mV dec$^{-1}$), and 40% Pt/C (32 mV dec$^{-1}$). (**c**) The polarization stability tests of Pt–MoS$_2$ between 0.13 and +0.57 V (vs. reversible hydrogen electrode (RHE)) at 100 mV s$^{-1}$ (solid line: Initial state; broken line: After 5000 cycles of CV scans) [44].

Wang and his group predicted the trends in HER activity of MoS$_2$ structures with different transition metals (Co, Fe, Ni, Cu) doping via a computational descriptor-based approach [45]. The enhancement in HER incorporating transition metal doping into the edge sites of the vertically aligned MoS$_2$ nanofilms was confirmed experimentally. Their polarization curves shows that Fe, Co, Ni, and Cu-doped MoS$_2$ nanofilms achieve $-2.3$, $-3.5$, $-2.4$, and $-2.6$ mA cm$^{-2}$ at the overpotential of 300 mV, respectively, around four times that of the pristine MoS$_2$ ($-0.6$ mA cm$^{-2}$ at 300 mV), indicating the large enhancement after the transition metal atoms incorporated into the edge sites. The Tafel plots corresponding to the above polarization plots suggest that the difference in Tafel slopes of pristine MoS$_2$ (118 mV dec$^{-1}$) and MoS$_2$ with doping (from 117 to 103 mV dec$^{-1}$) results is very small, indicating that the rate-limiting step is not changed after the doping process. These results show that MoS$_2$ could be doped with various transition metal dopants to discover new MoS$_2$-based catalysts in the future [45].

## 5. Surface Modification on 2D-Layered TMDs

Huang et al. reported that a composite film of molybdenum diselenide nanosheets (MoSe$_2$ NSs)/poly(3,4-ethylenedioxythiophene):poly(styrenesulfonate) (PEDOT:PSS) was fabricated using a low-cost drop-coating method for the counter electrode (CE) of a dye-sensitized solar cell (DSSC) [46]. The morphological features of various CEs are shown in Figure 11. The smoothest morphology of the bare PEDOT:PSS film is shown in Figure 11a. However, the smooth surface could lead to few electrocatalytic active sites and poor electrochemical surface. Compared with the bare PEDOT:PSS film, the composite films of MP-1.00 (MoSe$_2$ NSs in the PEDOT:PSS matrix) show rougher surfaces, as shown in Figure 11b. As shown in Figure 11c, the surface roughness of MoSe$_2$ NSs (planar size: 100–800 nm; thickness: 10–50 nm) in the bare MoSe$_2$ film is high. However, the poor adhesion of MoSe$_2$ NSs in the bare MoSe$_2$ film hinders electrocatalytic applications. With the incorporation of PEDOT:PSS, the composite films of MoSe$_2$/PEDOT:PSS would exhibit better electrocatalytic activity as compared to the bare MoSe$_2$ film; besides, the PEDOT:PSS could act as a binder to enhance the adhesion on the substrate, which could greatly improve the electro-catalytic ability and mechanic stability. Figure 12 shows the j–V curves of the DSSCs with the CEs of Pt, bare PEDOT:PSS, MP-1.00, and bare MoSe$_2$. The pertinent photovoltaic parameters are listed in Table 1. The power conversion efficiency (η) of the DSSC with a composite film having equal weights of MoSe$_2$ and PEDOT:PSS (MP-1.00) is 7.58 ± 0.05%, which is competitive with the efficiency of a Pt CE (7.81 ± 0.03%) [46].

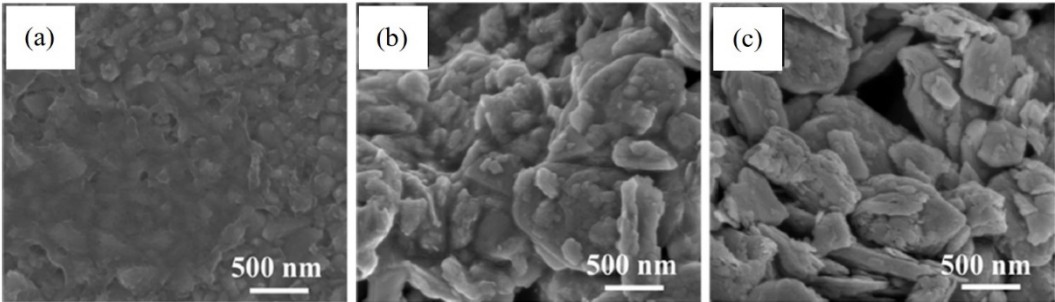

**Figure 11.** FE-SEM images show the surface morphology of (**a**) bare PEDOT:PSS, (**b**) MP-1.00, and (**c**) bare MoSe$_2$ [46].

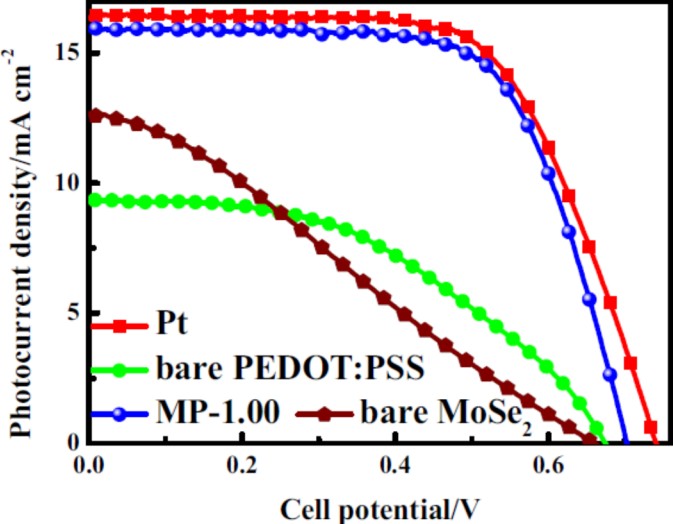

**Figure 12.** J–V curves of the (dye-sensitized solar cell) DSSCs with Pt counter electrodes (CEs), bare PEDOT:PSS CEs, MP-1.00 CEs, and bare MoSe$_2$ CEs [46].

**Table 1.** Photovoltaic parameters (η (%), V$_{oc}$ (V), j$_{sc}$ (mA cm$^{-2}$), and fill-factor (FF)) of the DSSCs with various CEs were measured at 100 mW cm$^{-2}$ (AM 1.5G). The standard deviation data for each kind of DSSC were obtained from three cells with the same fabrication processes [46].

| Counter Electrode | η (%) | V$_{OC}$ (V) | j$_{SC}$ (mA cm$^{-2}$) | FF |
|---|---|---|---|---|
| Pt | 7.81 ± 0.03 | 0.74 ± 0.00 | 16.38 ± 0.03 | 0.65 ± 0.00 |
| Bare PEDOT:PSS | 2.90 ± 0.03 | 0.67 ± 0.00 | 9.32 ± 0.25 | 0.46 ± 0.01 |
| MP-0.25 | 6.28 ± 0.25 | 0.67 ± 0.01 | 15.42 ± 0.55 | 0.61 ± 0.01 |
| MP-0.50 | 6.67 ± 0.15 | 0.68 ± 0.01 | 15.66 ± 0.29 | 0.61 ± 0.00 |
| MP-1.00 | 7.58 ± 0.05 | 0.70 ± 0.01 | 15.97 ± 0.18 | 0.67 ± 0.01 |
| MP-2.00 | 6.08 ± 0.04 | 0.68 ± 0.01 | 15.81 ± 0.21 | 0.57 ± 0.01 |
| Bare MoSe$_2$ | 2.29 ± 0.04 | 0.66 ± 0.01 | 12.65 ± 0.31 | 0.28 ± 0.01 |

Zhang et al. reported that the decoration of isolated Ni atoms on the surface of MoS$_2$ NSs supported on multichannel carbon matrix (MCM) nanofibers can improve the hydrogen evolution activity obviously [31]. As shown in Figure 13a, the hierarchical shell composed of randomly assembled ultrathin MoS$_2$ NSs can be clearly observed from the SEM image. There are no obvious changes in the original morphology of MCM@MoS$_2$ after the modification of the Ni atom (Figure 13b). In addition, the cross-sectional image of the multichannel structure of the MCM@MoS$_2$–Ni is shown in Figure 13c. As shown in Figure 14, compared with the original MoS$_2$ (−297 mV), MCM @ MoS$_2$ has a positively shifted overpotential at −10 mA cm$^{-2}$ (−263 mV), showing a significantly enhanced performance. Moreover, hydrogen evolution is considerably accelerated by the MCM@MoS$_2$–Ni

catalyst, and it shows a smaller onset overpotential ($-53$ mV) and higher current increase rate. This report represents a good method to activate the electrocatalytic activity of $MoS_2$ inert surfaces, which can be used in HER and other energy-related catalytic processes [31].

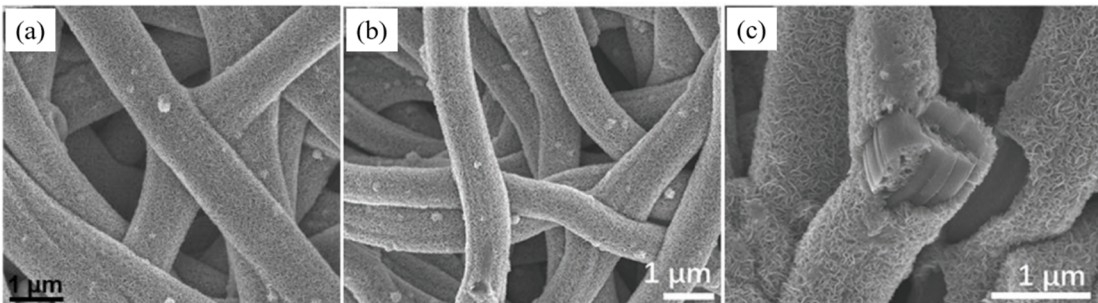

**Figure 13.** FESEM images of the multichannel carbon matrix (MCM)@$MoS_2$ nanofiber (**a**) without and (**b**) with Ni-atom modification. (**c**) The zoomed-in FESEM image of the hierarchical MCM@$MoS_2$–Ni nanofiber [31].

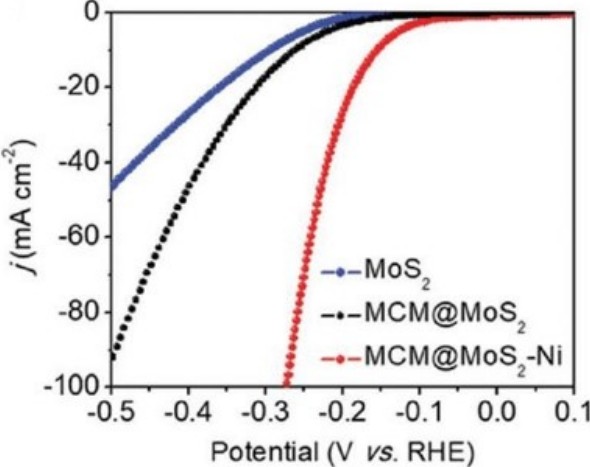

**Figure 14.** LSV curves of $MoS_2$ (blue line), MCM@$MoS_2$ (black line), and MCM@$MoS_2$-Ni (red line) in 0.5 m $H_2SO_4$ aqueous solution [31].

## 6. Edge Site Formation on 2D-Layered TMDs

In 2013, Xie et al. proposed creating active edge sites on the $MoS_2$ surface by the defect engineering for enhancing HER performance [30]. As shown in Figure 15, by designing a reaction with a high concentration of precursors and different amounts of thiourea, the modulation of defects on the surface of $MoS_2$ ultrathin NSs could be accomplished. As shown in Figure 16, the defect-rich $MoS_2$ ultrathin NSs showed a small onset potential of 120 mV, and a fast-rising current under higher negative potential. Compared with the highly crystalline samples (160–250 mV) and the nanostructured $MoS_2$, the defect-rich $MoS_2$ ultrathin NSs exhibited a good catalytic activity owing to the unique defect-rich structure, resulting in more active edge sites [36,47] By contrast, polarization curves of defect-free $MoS_2$ NSs and calcined $MoS_2$ NSs showed not only poor HER performance with high onset potential (180 and 230 mV) but also low cathodic current density. In addition, although the onset overpotential of the thicker NSs is competitive with the defect-rich $MoS_2$ ultrathin NSs, the cathodic current density is still unsatisfactory due to the limitation of the exposure of defect-induced active sites by assembly morphology. Therefore, the HER performance could be boosted significantly by the rich defects in the ultrathin NSs. As shown in Table 2, the defect-rich $MoS_2$ ultra-thin NSs show the highest active site density, proving the primary enrichment effect of active edge sites resulting from rich defects. Furthermore, the NS structure could produce more available defect-induced edge sites owing to the higher surface area of the ultrathin NSs. Generally speaking, the

great electrocatalytic behavior of the defect-rich $MoS_2$ ultrathin NSs is attributed to the enrichment effect of active edge sites induced by abundant defects and the synergic effect of ultrathin NS morphology. This study demonstrates controllable defect engineering in $MoS_2$ ultrathin NSs and opens avenues to develop other 2D-layered electrocatalysts (such as $WS_2$ and $MoSe_2$) [30].

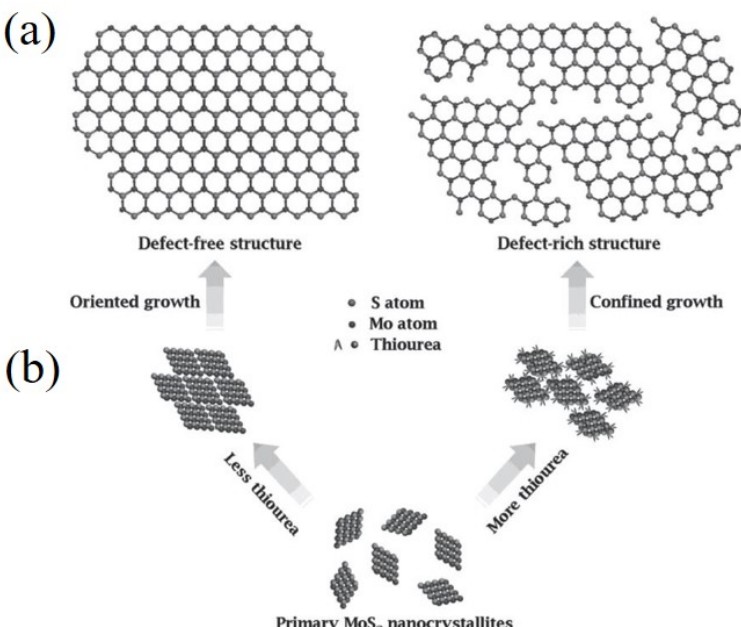

**Figure 15.** (**a**) Sketch of defect-free and defect-rich structures. (**b**) Synthetic processes of defect-free and defect-rich samples, respectively [30].

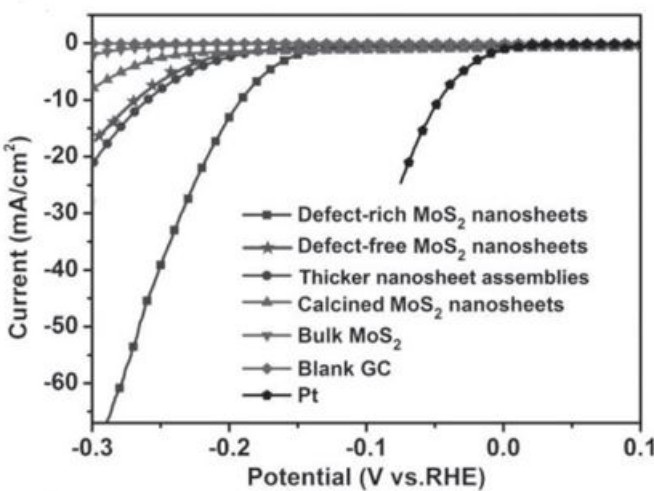

**Figure 16.** Polarization curves of various samples (defect-rich and defect-free $MoS_2$, thicker nanosheet (NS), calcined $MoS_2$, bulk $MoS_2$, blank GC, and Pt) [30].

**Table 2.** HER parameters (number of active sites ($10^{-3}$ mol g$^{-1}$), turnover frequency (TOF) (s$^{-1}$), Tafel slope (mV dec$^{-1}$), Tafel region (mV), $j_0$ (µA cm$^{-2}$), and $j$ (mA cm$^{-2}$)) of various MoS$_2$ samples [30].

| Materials [a] | Number of Active Sites [$10^{-3}$ mol g$^{-1}$] | TOF [s$^{-1}$] [b] | Tafel Slope [mV dec$^{-1}$] | Tafel Region [mV] | $j_0$ [µA cm$^{-2}$] [c] | $j$ [mA cm$^{-2}$] [d] |
|---|---|---|---|---|---|---|
| Defect-rich ultrathin NSs | 1.785 | 0.725 | 50 | 120–180 | 8.91 | 70.0 |
| Defect-free ultrathin NSs | 0.620 | 0.496 | 87 | 180–240 | 3.16 | 16.8 |
| Calcined NSs | 0.311 | 0.467 | 90 | 230–300 | 5.62 | 8.2 |
| Thicker NSs assembles | 0.582 | 0.653 | 88 | 120–250 | 7.94 | 20.3 |
| Bulk | 0.137 | 0.304 | 81 | 250–350 | 0.32 | 2.3 |

(a) All the parameters were measured under the same conditions, i.e., catalyst loading weight of 0.285 mg cm$^{-2}$ on glassy carbon electrode in 0.5 M H$_2$SO$_4$ aqueous solution; (b) TOFs were measured at η = 300 mV; (c) exchanged current densities ($j_0$) were obtained from Tafel curves by using extrapolation methods; (d) cathodic ($j$) was recorded at η = 300 mV.

Zhang et al. reported that the defect-rich MoS$_2$ monolayer with the two-dimensional reticulated structure was facilely prepared by a conventional hydrothermal method (surfactant ethyl xanthate (C$_2$H$_5$OC(=S)SNa) as sulfur source). To enhance electronic and photoelectric properties, the inert basal plane and delivered active sites in the MoS$_2$ monolayer should be triggered by increasing defects [48]. Figure 17 shows the Raman spectra of as-prepared MoS$_2$ (i.e., defect-rich MoS$_2$) and bulk MoS$_2$. The E$^1_{2g}$/A$_1$g peaks of defect-rich MoS$_2$ and bulk MoS$_2$ are at ∼380.7/406.1 and∼383.0/408.5 cm$^{-1}$, respectively. Compared to bulk MoS$_2$, the Raman peaks of defect-rich MoS$_2$ blue-shift, possibly owing to the vacancies in its structure [49]. Furthermore, the broader and weaker Raman peaks of the defect-rich MoS$_2$ than bulk MoS$_2$ imply its weak crystalline and abundant defects in its basal plane [50–52]. The S 2p$_{3/2}$ XPS spectra were deconvolved into two components, as shown in Table 3. The peaks at ∼161.33 and ∼162.75 eV correspond to the full-bonded S atoms of the MoS$_2$ monolayer and the part-coordinated S atoms, respectively [53–55]. Additionally, the atomic proportion of part-coordinated S atoms is 23.86%, estimating a lot of S vacancies in defect-rich MoS$_2$. There are two characteristic bands of Mo 3d$_{5/2}$ XPS at around 228.60 and 229.06 eV with a proportion of 57.90% and 42.10%, respectively, as shown in Table 3 [56]. The low band (228.60 eV) and the high band (229.06 eV) might be related to the Mo atoms with dangling bonds (lower valence than the full-coordinated Mo$^{+4}$ atoms) and the full-coordinated Mo atoms of MoS$_2$, respectively. Therefore, in addition to S vacancies, a lot of Mo vacancies exist in the defect-rich MoS$_2$. The HER activity of the defect-rich MoS$_2$ was measured by the electrochemical technique, as shown in Figure 18. The linear sweep voltammetry (LSV) curves in Figure 18a show the hydrogen evolution potentials of the defect-rich (235 mV) and bulk MoS$_2$ (707 mV), respectively. In order to reach the 10 mA cm$^{-2}$ current densities (j), the overpotentials (η) for the defect-rich and bulk MoS$_2$ should reach 244.18 mV with a 58 mV dec$^{-1}$ Tafel slope and 736.21 mV with a 173.6 mV dec$^{-1}$ Tafel slope, respectively, as shown in Figure 18b [57,58], indicating the superior catalytic activity and hydrogen evolution velocity of the defect-rich MoS$_2$. The electrochemical double-layer capacitance (C$_{dl}$) represents the active site quantities in the electrode interface, as shown in Figure 18c [59]. The larger C$_{dl}$ value and very small R$_{ct}$ value of the defect-rich MoS$_2$ imply the higher density of HER sites and high conductivity, respectively, possibly leading to the fast electron transfer and high-efficiency electrocatalytic process, as shown in Figure 18d [60]. The LSV curve for the initial cycle and that after 1000 CV cycles in Figure 18e showed similar results, implying the excellent stability of the defect-rich MoS$_2$ material. All above results reveal that the defect-rich MoS$_2$ possessed an excellent and stable activity for HER [48].

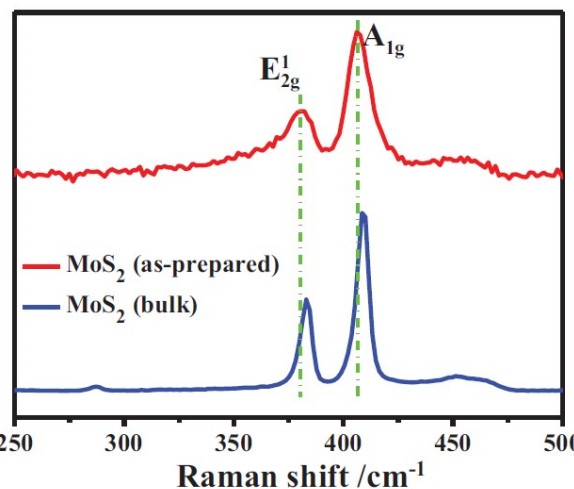

**Figure 17.** Raman characteristics of as-prepared (red line) and bulk MoS$_2$ (blue line) [48].

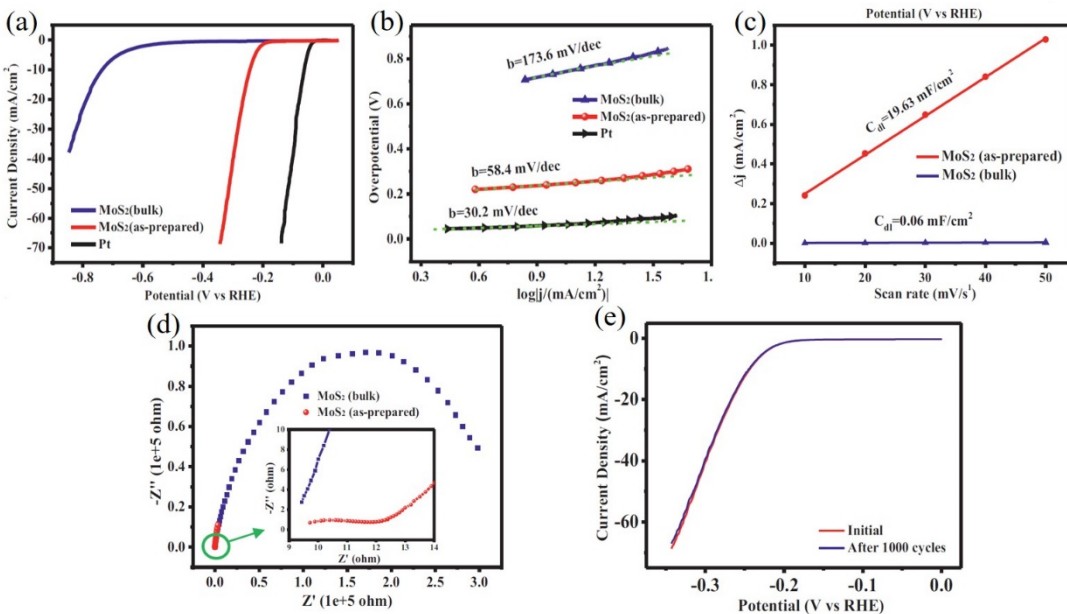

**Figure 18.** (**a**) Polarization curves and (**b**) Tafel slopes of bulk MoS$_2$ (blue line), as-prepared MoS$_2$ monolayer (red line), and Pt (black line), respectively. (**c**) The double-layer capacitances C$_{dl}$ of bulk MoS$_2$ (blue line), as-prepared MoS$_2$ monolayer (red line), (**d**) Nyquist diagrams of bulk MoS$_2$ (blue square), as-prepared MoS$_2$ monolayer (red dot) measured at 300 mV overpotential, and (**e**) the polarization durability test of the as-prepared MoS$_2$ monolayer (red line: Initial state, blue line: After 1000 cycles) [48].

**Table 3.** The S 2p and Mo 3d XPS peak parameters of the as-prepared MoS$_2$ monolayer [48].

| Elements | Binding Energy/eV | FWHM [c]/eV | Atomic Ratio/% | Affiliation |
|---|---|---|---|---|
| S (2p) | 161.33/162.51 [a] | 1.29 | 50.76/25.38 | S in MoS$_2$ |
| | 162.75/163.93 [a] | 1.30 | 15.74/8.12 | Vacant S in MoS$_2$ |
| Mo (3d) | 228.60/231.74 [b] | 1.25 | 25.61/16.49 | Mo in MoS$_2$ |
| | 229.06/232.39 [b] | 0.74 | 35.09/22.81 | Vacant Mo in MoS$_2$ |

[a] (S 2p$_{3/2}$)/(S 2p$_{1/2}$); [b] (Mo 3d$_{5/2}$)/(Mo 3d$_{3/2}$); [c] FWHM: Full-width at half-maximum.

## 7. Conclusions and Outlook

2D-layered TMDs possess promising catalysis properties and have been widely investigated for the applications on electrochemical energy. Especially, research on hydrogen evolution is theoretically well supported that 2D-layered $MoS_2$ exhibits much more favorable reaction coordinates for hydrogen evolution than most Pt-free catalysts [61]. In practically, though their basal plane shows poor activity toward catalysis reaction, several approaches, such as curvature engineering, plasma treatment, heteroatom-doping, edge site formation, and surface modification via nano-sized catalyst decoration, can be utilized to improve their catalysis activity efficiently. Future development of 2D-layered TMDs having superior catalytic activity would focus on the structural and interfacial engineering on 2D-layered TMDs. For example, the structural transformation of layered TMDs from a 2D sheet to 3D hollow sphere consisted of curved 2D TMDs, which could significantly increase the specific surface area and strain-induced active sites, as well as prevent the stacking and aggregation of most 2D-layered materials. Moreover, heteroatom-doping (e.g., single-atom doping or dual-atom co-doping) on curved 2D-layered materials is also theoretically demonstrated as a potential approach to further enhance the catalysis performance [62].

**Author Contributions:** Conceptualization and organization, C.-P.L. and D.-S.T.; Writing—original draft, C.-A.C., C.-L.L., and P.-K.Y. All authors have read and agreed to the published version of the manuscript.

**Funding:** This research received funding from Ministry of Science and Technology (MOST) of Taiwan.

**Acknowledgments:** This work was supported by the Ministry of Science and Technology (MOST) of Taiwan, under grant numbers 107-2113-M-845-001-MY3 and 109-2112-M-033-010-MY3.

**Conflicts of Interest:** The authors declare no conflict of interest.

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
