# Peer review of "Active Site Engineering on Two-Dimensional-Layered Transition Metal Dichalcogenides for Electrochemical Energy Applications: A Mini-Review"

_catalysts, doi:10.3390/catal11020151_

Round 1
Reviewer 1 Report
The manuscript entitled: “Active Site Engineering on Two-Dimmentional-Layered Transition Metal Dichalcogenides for Electrochemical Energy Applications: a Mini Review” submitted to Catalysts as Review describes modification of molibdemium disulfide or diselenide for different electrochemical energy application which requires electrocatalytical activity. Authors focused on presentation of different modification methods of thin layered molibdemium dichalcogenides and its influence on electrocatalytical properties, mainly parameters obtained during hydrogen evolution reactions in cyclic voltammograms. As complementary methods like TEM, SEM, Raman and photoluminescence spectroscopy. The manuscript describes influence of structural deformations, vacances and doping on material performance.
Study 2D transition metal structures is currently one of top trending topics in terms of replacement of expensive and scarce noble metals for different applications. In my opinion a review summarizing current state of the art . Presented manuscript provides an important information for the community and for this reason I recommend the manuscript for publishing with minor revision.
I would suggest to improve quality of reprinted figures. I would also suggest clearer presentation of reprinting note in the caption of figures.
Author Response
Reviewer 1:
General Comments:
The manuscript entitled: “Active Site Engineering on Two-Dimmentional-Layered Transition Metal Dichalcogenides for Electrochemical Energy Applications: a Mini Review” submitted to Catalysts as Review describes modification of molibdemium disulfide or diselenide for different electrochemical energy application which requires electrocatalytical activity. Authors focused on presentation of different modification methods of thin layered molibdemium dichalcogenides and its influence on electrocatalytical properties, mainly parameters obtained during hydrogen evolution reactions in cyclic voltammograms. As complementary methods like TEM, SEM, Raman and photoluminescence spectroscopy. The manuscript describes influence of structural deformations, vacances and doping on material performance. Study 2D transition metal structures is currently one of top trending topics in terms of replacement of expensive and scarce noble metals for different applications. In my opinion a review summarizing current state of the art. Presented manuscript provides an important information for the community and for this reason I recommend the manuscript for publishing with minor revision.
Response: We are thankful to Reviewer 1 for his/her positive comments. We have taken this reviewer’s precious comments seriously and revised the manuscript accordingly.
Specific comments/suggestions: 1. I would suggest to improve quality of reprinted figures. I would also suggest clearer presentation of reprinting note in the caption of figures.
Response: We thank the reviewer for this suggestion. In the revised manuscript, the quality of all reprinted figures are improved; moreover, all the caption of figures and tables are rewritten properly.
Reviewer 2 Report
The manuscript "Active Site Engineering on Two-Dimensional-Layered Transition Metal Dichalcogenides for Electrochemical Energy Applications: A Mini-Review" summarizes various efforts on improving the electrochemical device performance of TMDCs through active site engineering. Due to the following comments, this manuscript is recommended to be published after some revision
1. Overall image resolution should be improved. The original publishers of the cited papers should be providing high-resolution images. Please use that high-resolution image for this manuscript.
2. Please unify all the terminology used throughout the manuscript. For example, "TMDs" in the abstract and "TMDCs" in the manuscript should be unified. "MoS2 or MoS2" and "molybdenum diselenide" should be unified. Please go through the manuscript carefully.
3. For O2 plasma treatment, the authors mentioned "oxygen species doping" for better device performance, but at the same time, oxidation of MoS2 can happen to probably cause negative effects on the device performance. Please provide a clear description of possible disadvantages of the oxidation of MoS2 in the aspect of device performance.
Author Response
General Comments:
The manuscript "Active Site Engineering on Two-Dimensional-Layered Transition Metal Dichalcogenides for Electrochemical Energy Applications: A Mini-Review" summarizes various efforts on improving the electrochemical device performance of TMDCs through active site engineering. Due to the following comments, this manuscript is recommended to be published after some revision.
Response:
We are thankful to this reviewer (Reviewer 2) for his/her useful comments. We have taken the reviewer’s comments seriously and revised the manuscript accordingly.
Specific comments/suggestions:
- Overall image resolution should be improved. The original publishers of the cited papers should be providing high-resolution images. Please use that high-resolution image for this manuscript.
Response:
We thank the reviewer for this suggestion. In the revised manuscript, the quality of all reprinted figures are improved
- Please unify all the terminology used throughout the manuscript. For example, "TMDs" in the abstract and "TMDCs" in the manuscript should be unified. "MoS2 or MoS2" and "molybdenum diselenide" should be unified. Please go through the manuscript carefully.
Response:
We thank the reviewer for this comment. In the revised manuscript, we have replaced the original term with the right term as following, and changes are highlighted with yellow background.
“TMDCs”->“TMDs”
“2D materials”-> “2D-layered materials”
“MoSe2 nanosheets”-> “MoSe2 NSs”
“MoS2 nanosheets”-> “MoS2 NSs”
“nanosheets”-> “NSs”
- For O2 plasma treatment, the authors mentioned "oxygen species doping" for better device performance, but at the same time, oxidation of MoS2 can happen to probably cause negative effects on the device performance. Please provide a clear description of possible disadvantages of the oxidation of MoS2 in the aspect of device performance.
Response:
We thank the reviewer for this comment. In the revised manuscript, we have provided a description of possible disadvantages of the over oxidation of MoS2. (page 6, lines 3-7)
Reviewer 3 Report
This article is a comprehensive review on metal dichalcogenides for electrochemical applications and catalysis. It is well-written, contains a good summary on the structure and reactivity of these materials and it should appeal to a broad audience. I recommend its publication, with the request that the following key references on the synthesis and properties be cited:
1. Nature Chem 5, 263–275 (2013)
2. Nat Commun 6, 8063 (2015)
3. Sci Rep 5, 15718 (2015)
4. Nat Commun 11, 5032 (2020)
5. Nature chemistry, 7(1), 45 (2015)
6. Sci Rep 4, 5348 (2014)
7. Advanced Energy Materials, 7(23), p.1700571 (2017)
No further revisions are requested.
Author Response
General Comments:
This article is a comprehensive review on metal dichalcogenides for electrochemical applications and catalysis. It is well-written, contains a good summary on the structure and reactivity of these materials and it should appeal to a broad audience.
Response:
We are thankful to this reviewer (Reviewer 3) for his/her positive comments.
Specific comments/suggestions:
I recommend its publication, with the request that the following key references on the synthesis and properties be cited:
- Nature Chem5, 263–275 (2013)
- Nat Commun6, 8063 (2015)
- Sci Rep5, 15718 (2015)
- Nat Commun11, 5032 (2020)
- Nature chemistry, 7(1), 45 (2015)
- Sci Rep4, 5348 (2014)
- Advanced Energy Materials, 7(23), p.1700571 (2017)
No further revisions are requested.
Response:
We thank the reviewer for this suggestion. Above key references have been cited in the revised manuscript, as shown in the “Reference Section” highlighted with yellow color.